# “Are You Just Looking to ‘Survive’?”: A Qualitative Study of Importance of Oncology Endpoints Beyond Overall Survival in Early-Stage Cancer [note 1]

**DOI:** 10.3390/cancers17193260

**Published:** 2025-10-08

**Authors:** Shelagh M. Szabo, Sarah Walker, Evelyn Griffin, Aya McMillan, Robert Bick, Frances Simbulan, Eon Ting, Stephanie Snow

**Affiliations:** 1Broadstreet HEOR, 300–177 West 7th Ave, Vancouver, BC V5Y 1L8, Canada; 2Collaborator at Rethink Breast Cancer, 50 Carroll St, Toronto, ON M4M 3G3, Canada; 3CanCertainty Coalition, 3-251 Queen Street South, Suite 561, Mississauga, ON L5M 1L7, Canada; 4AstraZeneca Canada, 1004 Middlegate Road, Mississauga, ON L4Y 1M4, Canada; 5QEII Health Sciences Centre, Dalhousie University, Halifax, NS B3H 2Y9, Canada

**Keywords:** qualitative research, patient-centered care, patient outcome assessment, medical oncology

## Abstract

This study explored how people treated for early-stage or curable cancer in Canada view outcomes measured in clinical trials. Traditionally, trials focus on overall survival (how long people live), but newer studies often use other outcomes, such as being free of cancer returning (recurrence-free survival), staying free from signs that the cancer is worsening (event-free survival), or showing no cancer after treatment (pathological complete response). Thirty-three people treated for breast, lung, or gastrointestinal cancers were interviewed. Participants said living longer was important, but they also valued outcomes that reflect living without cancer and being able to return to normal life. Many emphasized the importance of timely and fair access to new treatments, even when results are based on these non-survival outcomes. Overall, the study shows that patients see these measures as meaningful and support using them to speed up access to promising treatments.

## 1. Introduction

Despite advancements in screening, diagnosis, and treatments, which have substantially improved survival, cancer remains a leading cause of death worldwide [1]. For individuals with early-stage cancer, treatments vary based on stage and type, biomarker status, and clinical factors [2]. The treatment landscape for early-stage disease across most cancer types has been rapidly evolving, with increasing evidence supporting the use of neoadjuvant and/or adjuvant treatments to reduce the risk of recurrence, thereby improving long-term survival [3,4,5,6,7].

New treatments in oncology have traditionally been evaluated by improvements in overall survival (OS), “the time from randomization until death from any cause”, providing a standardized measure of efficacy for comparison across trials [8]. As an outcome measure, OS is understood by clinicians and patients, and accepted by regulatory and health technology assessment (HTA) decision-makers. However, many trials employing OS have been conducted in advanced, metastatic settings where OS may be observed over a shorter trial period, and there may be substantial challenges with assessing OS as the primary outcome, which has been garnering increasing attention [8,9,10]. For example, in early-stage cancer trials, patients are often treated with curative intent, and observing OS over many years can be infeasible. Estimates of OS in early-stage cancer may be confounded by the efficacy of subsequent treatments, the risk of relapses or progression, and other comorbidities [11]. Relying on OS data for regulatory and reimbursement approval could delay access to effective and potentially lifesaving therapies [8,9,10]. Further, in both early- and advanced-stage cancers, there is an increased recognition of the inherent value of aspects not directly measured by OS, especially from the perspective of patients and their caregivers. These include impact on health-related quality of life (HRQoL), short- and long-term treatment-related adverse effects, and level of physical functioning, knowledge of which would require assessment of alternative endpoints [11,12,13].

Due to the limitations of relying solely on OS data for establishing therapeutic benefit [10], endpoints assessed at earlier time points are being considered for decision-making [8,10,14]. These ‘non-OS’ endpoints—which include pathologic complete response (pCR), recurrence-free survival (RFS), disease-free survival (DFS) or event-free survival (EFS)—enable timely measures of treatment efficacy in advance of mature OS data [8,13,14,15]. While there is some recognition of the value of non-OS endpoints for some tumor types [8,16], non-OS endpoints are not yet widely accepted by HTA bodies for quantifying treatment value. This reflects the reality HTA bodies and payers face regarding the ‘OS impasse’ and how to weigh potential benefits of new therapies against financial considerations and established value metrics [10,17]. Given the rapid advancements in new treatments for early-stage cancers, it is also important to understand the perspectives of people with cancer on the value of non-OS endpoints, as endpoints used in clinical trials should be patient-relevant and meaningful [18]. Previous quantitative research found that cancer patients consider pCR, DFS and OS as distinct and relevant endpoints [19]. Qualitative assessments of non-OS endpoints, which can be useful to understand the context and significance patients attach to these [20], are limited [21]. An understanding of the perceived value and limitations of these endpoints from a patient perspective is important in order to ensure that assessments of the value of new cancer therapies reflect the priorities of patients and their families [22]. The objective of this study was to explore the perspectives of Canadians with early-stage cancer on the perceived value of non-OS endpoints.

## 2. Materials and Methods

### 2.1. Study Design

In this cross-sectional study, we recruited Canadian residents aged > 18 years who were currently receiving treatment or were previously treated with curative intent for the following cancer types and stages: stage I-IIIC breast cancer, I-IIIC non-small-cell lung cancer, limited-stage small-cell lung cancer, or gastrointestinal (GI) cancer (stage I-IIIB esophageal, I-IIIC gastric, or stage 0/A/B liver cancer). These stages were selected according to each type of cancer to represent cancer of early stage or treatable with curative intent and were intentionally selected to include stages beyond stage 1 cancer. Participants were deemed ineligible if they did not speak fluent English, were not able to provide informed consent, or were diagnosed with other GI cancers outside of those listed previously (e.g., colorectal cancer).

### 2.2. Sample Size and Recruitment Criteria

The target sample size was 30–35 participants to allow diversity in cancer type, stage, and treatments received (relevant to treatment with curative intent or for individuals considered to have early-stage cancer). The sample size was informed by published guidelines for achieving information saturation in qualitative interviews and prior study design [21,23,24,25,26]. We purposively aimed to recruit 10–12 individuals diagnosed with breast cancer, 10–12 with lung cancer, and 5–10 with GI cancer from across Canada.

### 2.3. Participant Recruitment

Participants were recruited through CanCertainty, a coalition of Canadian cancer patient groups, charities, and caregiver organizations that work with cancer care professionals to improve the affordability and accessibility of cancer treatment. Recruitment invitations were sent to the CanCertainty network through distribution lists. For harder to recruit segments (e.g., those with GI cancer), recruitment was ultimately supplemented via the patient databases of a recruitment partner, Global Perspectives.

Potential participants were provided with details of the study objectives and protocol. During screening, participants’ cancer type and stage were ascertained; no relationship between participants and interviewers existed prior to the interview.

### 2.4. Data Collection

Semi-structured interview guides were developed in collaboration with a medical oncologist (SS) and a patient expert partner with experience with breast cancer (AM). Endpoint definitions were developed from the published literature [8,13,14,15,27,28,29], adapted for a lay audience, illustrated with explanatory graphics (Appendix A; Figure A1), and validated by our clinical and patient expert collaborators. All study materials were pilot tested with three members of the general population and our patient collaborator.

At the beginning of the one-on-one virtual interviews, the explanatory graphics were shared on screen. Interviewers used these graphics to explain the clinical endpoints being discussed, allowing time to ask questions and clarify understanding before proceeding. Participants then discussed their treatment and provided perspectives on OS and non-OS endpoints; including how well endpoints aligned with their treatment priorities, and their thoughts on the implications of evaluating a new therapy based on non-OS endpoints. Participant clinical history and demographics were also collected. Interviews were conducted by three researchers (SMS, EG and SW) with qualitative training. To the best of our knowledge, no one was present aside from the interviewer and participant at each interview. Key points within the interview were summarized by the interviewer throughout the interview; further member checking was not performed [30].

As interviews progressed, materials were iteratively updated based on participant feedback, and interviewers met regularly to discuss any challenges and promote consistency. Incoming data were regularly reviewed, and recruitment continued until saturation was judged to be reached. This was defined as no new strengths or drawbacks to non-OS endpoints identified after five consecutive interviews. Interviews lasted approximately one hour and were audio-recorded. Informed consent was obtained from all subjects involved in the study. All participants received remuneration.

### 2.5. Analyses

Interviews were transcribed; transcripts were independently reviewed by two analysts for data familiarization. Reflexive thematic analysis was used to explore patterns in responses [31]. Codes were assigned to describe treatment experiences, views on endpoints, and alignment of endpoints with treatment goals, priorities, and preferences. Coding was iterative, with continuous addition of new codes and refinement of existing codes. Codes were grouped into categories that formed the basis for constructed themes. Supportive quotations were identified from transcripts, and themes were visualized using a thematic map representing the underlying coding tree.

The qualitative analysis was conducted by two researchers (SMS and SW), collaborating with the broader team for theme development. The analysis was performed iteratively alongside rounds of data collection to help identify topics requiring further investigation and whether further recruitment was necessary.

The clinical and sociodemographic characteristics of the sample were summarized.

Coding for reflexive thematic analysis was performed using NVivo 14 (Lumivero, Denver, CO, USA), and Microsoft Excel (Microsoft Corporation, Redmond, WA, USA) was used to tabulate quotations and quantitative analyses.

### 2.6. Reflexivity and Ethics

We acknowledge that our own likely treatment goals if confronted with cancer could have influenced our interpretations of participant statements. Conversely, given that we largely do not have personal cancer experience, there may be perspectives overlooked. This gap in personal insight was addressed by including a patient partner and an oncologist on the study team. As many of the research team work in health economics, our understanding of how clinical trial data are used by decision-makers may have resulted in us over-interpreting the words of participants, attributing to them a depth of understanding that would be unlikely for someone with lesser familiarity.

The authors assert that all procedures contributing to this work comply with the ethical standards of the relevant national and institutional committees on human experimentation and with the Helsinki Declaration of 1975, as revised in 2013. Ethical approval was obtained from the WCG IRB in January 2025 (file number: 1370874), and study reporting guided by the COREQ (COnsolidated criteria for REporting Qualitative research) checklist (Appendix B).

## 3. Results

### 3.1. Demographics and Clinical Characteristics

Of thirty-three participants, twelve reported a diagnosis of breast cancer, eleven of lung cancer, and ten of GI cancer; mean age was 54.8 (standard deviation = 12.9) years, and twenty-one were female (Table 1). No individuals withdrew after providing consent. Eleven participants were receiving treatment at the time of the interview, including five with recurrence. Twenty-eight participants had undergone surgery, twenty-six received chemotherapy, and twenty-one were treated with both (ten specified neo-adjuvant and nine adjuvant). Eighteen participants had no evidence of disease at the interview (Table 1).

### 3.2. Participant Perspectives on OS as a Trial Endpoint

All participants highly valued survival, noting several strengths to using OS as an endpoint (Table 2). They found the concept of OS easy to understand, and acknowledged it provides a benchmark for understanding how different therapies perform. They often regarded survival as a principal goal of treatment:

“It helps you plan for the rest of your life. If somebody says to you, ‘You have one year to be alive’ or ‘You have 10 years to be alive’ I think you would behave differently. You would pack these 10 years in one year…[OS] has some importance for the quality of life and what you want to achieve in your life”—ID11

However, participants also noted that OS alone does not capture all relevant aspects of their experiences. Other noted limitations included length of time needed to measure OS; that OS estimates can be misleading in early-stage cancers because of the inability to account for competing mortality risks; and that details of prognosis based on population averages may have limited personal relevance. Many participants expressed that OS does not explicitly consider HRQoL, focusing instead on length, and noted survival without good HRQoL was not desirable by most.

“It’s a funny word, ‘survival,’ because that … captures the essence of everything you’re asking, really. Are you just looking to ‘survive’? Are you looking to really lead a completely normal and healthy life going forward, and where is the gap between those two things?”—ID26

### 3.3. Participant Perspectives on Non-OS Endpoints

Participants felt non-OS endpoints offered value (Figure 1; Table 3), as they reflect their treatment goals and priorities. These included maintaining HRQoL and avoiding side effects; getting back to ‘normal’ after treatment by being disease-free; stopping cancer progression; and avoiding relapse over a longer term. Participants believed the information conveyed by non-OS endpoints provided additional clinical details beyond survival, and that knowledge of being cancer-free provides hope. As such, they felt clinical benefits demonstrated by early endpoints are sufficiently compelling to want access to new treatments (Table 4).

Participants placed high value on timely and equitable access to treatments, feeling this imperative to saving lives (Figure 1; Table 3). They equated having more treatment options with having better options, and desired access to cutting-edge treatments that better meet treatment goals (Table 4). Participants understood that treatments for early-stage cancer would take longer to be approved if OS was the primary endpoint, which would result in delayed access, and they expressed concerns that inequitable access could arise if treatments were available in certain places but not others:

“I can’t imagine them waiting for overall survival to put something on the market. Then I find it amazing how many people are saying ‘‘Well, the drug is available in the States, but it’s not available here yet,’ or…It’s not covered by our insurance yet, but it’s covered by so-and-so’s insurance.’… I don’t think they can wait for [OS]. They just have to do it. You have to save as many of us as possible.”–ID22

However, participants did identify some drawbacks to the use of non-OS endpoints (Figure 1; Table 3). Some noted that treatments measured to the standards of non-OS endpoints may not meet the expectations of all patients, and they still felt these endpoints did not capture all relevant aspects to decision-making. Finally, there was some concern that use of non-OS endpoints may compromise research transparency, and there would be less available information on long-term clinical benefits and side effects of the new treatment.

### 3.4. Valuing Specific Non-OS Endpoints Differently

Participants did not obviously distinguish between RFS, DFS, and EFS in terms of importance (Table 4). When asked about RFS or DFS, participants articulated the value of knowing disease status or the potential for a recurrence or new cancer to occur. However, EFS was more difficult for participants to conceptualize given the varied outcomes that the definition could encompass.

Participants generally agreed that achieving pCR was a priority in their own treatment journey, providing important information conveying hope for future treatment success, or peace of mind if they had no evidence of disease. For some (e.g., those undergoing mastectomy or lobectomy for lung cancer), having pCR identified on their surgical pathology was a principal goal. These individuals valued pCR as reassurance towards being disease-free in the short term rather than focusing on longer-term survival. Participants believed that achieving pCR would lead to extended RFS, DFS or EFS, and that clinical benefits in these endpoints would result in longer OS—even when informed that evidence for these associations is presently limited. The sense of hope and reassurance associated with achieving pCR or prolonged DFS or RFS appears to be an important part of how participants valued non-OS endpoints. Nonetheless, participants found pCR as an endpoint within a clinical trial context more difficult to conceptualize. As a result, the cohort predominantly favored medium- to long-term outcomes such as maintaining HRQoL and remaining cancer-free that are reflected in endpoints like RFS, DFS and EFS.

“[Being] recurrence-free would be more aligned with what I would want, because obviously the goal would be not to have to go through chemo, radiation, [and] surgeries again. So, of course, overall survival is important, but if I’m living but I’m living with other cancer and going through treatment, that wouldn’t be the goal.”—ID05

### 3.5. Other Factors Influencing Perspectives of Importance of Endpoints

Across tumor types, the relative importance of being disease- or recurrence-free versus maximizing length of life differed based on age, recurrence status, prognosis, life stage, and treatment history (Table 4).

Specific to age, younger participants and/or those with better prognoses—which included many with breast or liver cancer—typically valued having more and better treatment options. Many exercised choice in their treatment, often selecting options ensuring physical tumor removal. These individuals often expressed that their main goal was to be disease-free, and a corresponding preference for non-OS endpoints.

“I was hoping to be cured, of course. That was my main goal. If not, something that would extend my life for another few years. The other one, of course, was to be able to continue on with my life and return to work and spend time with my family.”—ID14

In contrast, those diagnosed at older ages, who had relapsed or continued to have evidence of disease (including many participants with lung or non-liver GI cancer) often described changing perspectives since diagnosis. For these participants, survival was important but being cured may not still be an option. This reinforces the importance of considering treatment impact beyond OS as maintaining HRQoL in the need for potentially longstanding cancer management became a more important personal treatment goal. As many of these individuals had progressed through multiple therapies and were worried about exhausting treatment options, they placed high importance on therapeutic innovation. As highlighted by a participant for whom the only option after surgery was a clinical trial, many would try almost anything to achieve the best possible outcome:

“After my surgery, when the clinical trial was offered, I was told that the immunotherapy drug had not been tried on my type of cancer. So, I was the first one to get it. Throwing something at the wall and seeing if it would stick. But it was all that was being offered … so I jumped at it.”—ID36

### 3.6. Patient Agency and Decision-Making at an Individual and Population Level

Some participants desiring a more active role in decision-making noted they do not always feel empowered to advocate for their own treatment goals, particularly when these differ from those of their care teams or emphasize outcomes other than survival. These participants expressed their desire for clinicians to consider the needs of patients as individuals when providing information to guide treatment decision-making. Some felt they had received insufficient information on the risk of recurrence or HRQoL impact when starting therapy—information that may have been conveyed by data on RFS, DFS or EFS—that resulted in them making a treatment decision they might not have had they had more information (Table 4).

Participants expressed a desire for decision-makers to consider access to treatments for early-stage cancers based on non-OS endpoints. Many, but not all, participants would value the opportunity to consider treatments with efficacy expressed in non-OS endpoints, even in the absence of OS data. Conversely, participants recognized the need to balance considerations around treatment novelty and shorter-term evidence of efficacy, with considerations around costs, side effects, and the lack of proven long-term survival benefit associated with non-OS endpoints. Even among those who felt non-OS endpoints strongly aligned with their treatment values, they remained concerned about side effects that “could be worse than the disease” or could negatively impact their lifestyle.

Finally, participants were encouraged by timely scientific research, particularly in situations where they felt existing treatment options were insufficient. Many expressed interest in joining clinical trials if it meant they could receive cutting-edge care without waiting for a lengthy drug-approval process. Particularly for those who had a recurrence, never attained ‘no evidence of disease’, or had run out of treatment options, they saw scientific innovation as a direct path to survival. Nonetheless, participants had different perspectives on how innovative research should intersect with their own treatment options. Novel treatments were not necessarily ‘right’ for all individuals, but even those not personally interested in innovative treatments felt that patients should have the opportunity to consider them, should the results for clinical benefits align with their treatment goals.

“It’s really encouraging to know that alternative endpoints are being considered that can provide patients with greater options for therapy … Whether it’s a complete response or a cure or just maintaining quality of life, all of those things are important. … My guess is that in talking to cancer patients, you will be hearing a lot of the same feedback, that people are willing to take those risks because the alternative is worse, and that if people are willing to try that with the guidance of their medical teams, those options should be available.”—ID02

## 4. Discussion

The treatment landscape for early-stage cancers continues to advance, and strategies to support the timely evaluation of treatment benefits, including considering endpoints beyond OS, are needed [10]. In addition to needing buy-in from decision-makers on these endpoints, they also should be patient-relevant and meaningful, ensuring that outcomes measured in clinical trials matter to patients and their families [33]. Our research highlights that, from the perspective of Canadians with early-stage cancer, non-OS endpoints like RFS, DFS or EFS are personally meaningful and highly related to their treatment priorities. These findings extend on a previous qualitative study reporting that lung cancer survivors found DFS to be intrinsically meaningful to treatment outcomes and aligned with patient goals, believing that longer DFS would help maintain better HRQoL [21]. Another study reported that liver cancer patients noted that OS does not take into account HRQoL, which they highlighted is an important concept when discussing the life expectancy of patients with cancer [34]. While these sentiments were echoed by our study participants, we also noted a moderating effect of age and life stage across cancer types; potentially due to the larger sample size and diversity of patient experiences within our sample.

The study findings also complement those of a discrete choice experiment (DCE) by Batchelder et al., reporting that early-stage breast cancer patients had the highest preference for pCR, followed by OS and DFS [35]. Another mixed-methods study reported that pCR was a patient-relevant outcome for neoadjuvant therapy in breast cancer, valued more highly than OS or DFS [19]. Those authors theorized that the importance placed on pCR could be due to participants’ assuming their response to therapy would translate to the destruction of circulating cells and result in them achieving a cancer-free status [19]. Participants in our study also valued the hope that attaining pCR could provide. However, they also considered this hope to be a benefit provided by RFS, DFS, and EFS as well; using language such as ‘getting back to normal’ and ‘kick [ing] it to the curb’ to relate the importance of knowing they were disease-free. Overall, our qualitative findings provide a deeper understanding of the reasons that people prefer different non-OS endpoints, and in which situations. For example, while participants found pCR relevant to their own personal experience, when considering the clinical utility of the non-OS endpoints, they valued RFS, DFS and EFS for the longer-term information on HRQoL and recurrence they believed these endpoints provide.

Batchelder et al. also noted that endpoint importance differed according to age and that breast cancer patients with more recent diagnoses placed more relative importance on pCR vs. those with longer times since diagnosis valuing OS [35]. We also identified variability in age, prognosis, and treatment status impacts the perceived importance of being cancer-free. Taken together with the present study, these findings underscore the importance of patient characteristics when considering which endpoints are most relevant.

Achieving personal treatment goals is not the only reason that participants valued non-OS endpoints—broader treatment access issues were also a priority. A recent US-based DCE reported that participants with cancer were willing to wait for a potential treatment with a survival benefit (vs. an existing treatment with low certainty of survival benefit); conversely, they also valued faster access to new treatments [36]. We also noted that participants valued the long-term safety and efficacy information that the OS endpoint can provide. Nonetheless, most participants were very concerned about timely access to innovative treatment, which they understood could be facilitated through using non-OS endpoints in decision-making [13]. These conflicting viewpoints may be partly explained by the subset of participants with lower tolerance of risk for novel treatments, but who nonetheless support their development and more widespread access.

Participants also highlighted their interest in the information conveyed by non-OS endpoints on HRQoL impact and risk of recurrence. A drawback to the current focus on OS to evaluate treatment efficacy is that it may make it challenging for patients to understand information provided on less familiar non-OS outcomes, and further education around non-OS endpoints will help ensure patients can make informed treatment decisions. This gap has been documented in a mixed-methods study that noted that, although information on clinical endpoints (OS, progression-free survival, response rate) was readily available online, endpoint definitions relied on technical language that may not be easily understood. In focus groups, few participants demonstrated a deep understanding of these concepts even after the endpoints were explained; for example, many assumed that all endpoints could indicate whether a cure was achieved or length of life extended [34]. These findings underscore the complexity of communicating clinical outcomes to patients and raise important considerations with how to balance patient desires for novel treatments with expectations for evidence-based decision-making that also considers economic value. Patient input pathways for HTA can help ensure rigorous patient-derived evidence informs decisions being made [37,38], and deliberative processes facilitating continued evaluation of evolving evidence, such as those proposed for precision oncology medicine, may also offer a path forward [39].

In this study, we focused specifically on patients with esophageal, gastric, and liver subtypes of GI cancer for two reasons. Firstly, compared to colorectal cancer, which has a substantial body of psychosocial literature, esophageal, gastric, and liver cancers remain under-represented in qualitative research [40,41,42]. Secondly, these cancers present unique clinical and experiential challenges in the curative setting, including complex multimodal treatments and uncertainties around long-term prognosis [42,43]. Similarly, we focused not only on patients with stage I (early stage cancer) but also those with later stage cancers who were treated with curative intent, to collectively explore the preferences and priorities of Canadians navigating treatment journeys most relevant to outcomes being documented using non-OS endpoints.

The robust qualitative methods used enabled a deep exploration of the perspectives driving patient preferences, complementing available quantitative preference data [44]. Additional strengths include the diversity of experience and perspectives offered by including participants with different cancer types and disease stages (among those treated with curative intent). While strata were set for key clinical characteristics, recruitment targets for characteristics such as province, cultural background, or educational attainment were not. As a result, this study has limited representation of ethnic minorities, lung cancer participants were mostly female despite a higher incidence in males, and approximately one-third held postgraduate degrees. This represents a limitation to generalizability as those with higher educational attainment may have different health literacy, access to care, or attitudes towards and awareness of the outcomes discussed. Although conducting research online may reduce barriers to participation, it may exclude those who are less comfortable with online technology [45]. Finally, disease status was self-reported and not corroborated clinically.

This study highlights several important directions for future work. First, understanding HTA body perspectives on this research and non-OS endpoints in general would be valuable as a next step to help ensure trials are measuring outcomes that are meaningful to patients while also providing the necessary data for understanding treatment value. Additionally, extending this research to incorporate the perspectives of lesser-represented or marginalized groups, and including people with a broader variety of cancers will help promote equity and ensure that patient-centered evidence is representative and inclusive.

## 5. Conclusions

This study used reflexive thematic analysis to explore the importance of cancer trial endpoints beyond OS for individuals treated for early-stage cancers or with curative intent, through the lens of their own treatment goals and priorities. Participants highlighted the value they perceived the non-OS endpoints to have due to their alignment with treatment goals, such as being ‘cured’, minimizing side effects, avoiding recurrence, and maintaining quality of life. Participants also valued the clinical details those endpoints convey. Participants were invested in access to novel treatments with meaningful improvements in HRQoL and non-OS endpoints, even in the absence of OS data. These findings provide evidence of the patient-perceived value of non-OS oncology endpoints to support decision-making, and highlight the impact of patient context and experiences on treatment goals and endpoint preferences.

## Figures and Tables

**Figure 1 cancers-17-03260-f001:**
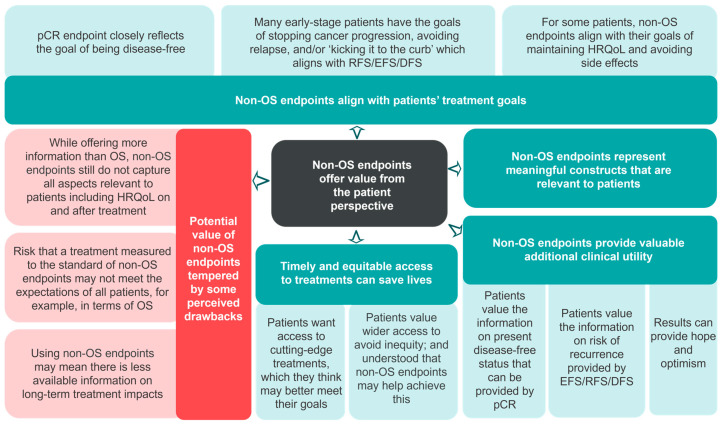
Perceived strengths (teal) and drawbacks (red) to the use of non-OS endpoints. Abbreviations: DFS, disease-free survival; EFS, event-free survival; HRQoL, health-related quality of life; OS, overall survival; pCR, pathological complete response; RFS, recurrence-free survival.

**Table 1 cancers-17-03260-t001:** Participant demographic and clinical characteristics, for the overall sample (*n* = 33), stratified by type of cancer.

			Diagnosed with:	
	Overall (*n* = 33)	Breast Cancer (*n* = 12)	Lung Cancer (*n* = 11)	GI Cancer ^3^ (*n* = 10)
Age at interview, mean (SD) yrs	54.8 (12.9)	46.3 (9.3)	62.9 (8.9)	56.5 (18.2)
Female, *n* (%)	21 (64%)	12 (100%)	8 (72%)	1 (10%)
Age at diagnosis, mean (SD) yrs	50.4 (12.4)	41.7 (8.5)	57.3 (9.8)	54.0 (13.7)
Disease stage; Non-liver cancers/liver cancer, *n* (%)				
I/0	10 (30%)	6 (50%)	2 (18%)	2 (20%)
II/A	12 (36%)	3 (25%)	2 (18%)	7 (70%)
III/B	7 (21%)	1 (8%)	5 (45%)	1 (10%)
Unknown	4 (12%)	2 (17%)	2 (18%)	-
Time from diagnosis to treatment, *n* (%)				
1 month or less	16 (49%)	8 (67%)	3 (27%)	5 (50%)
2–3 months	8 (24%)	3 (25%)	2 (18%)	3 (30%)
4–5 months	3 (9%)	-	3 (27%)	-
More than 6 months	1 (3%)	-	1 (9%)	-
Recurrence status, *n* (%)				
No evidence of disease	18 (55%)	6 (50%)	8 (73%)	4 (40%)
Treatments received, *n* (%)				
Surgery	28 (85%)	12 (100%)	8 (73%)	8 (80%)
Surgery alone	3 (9%)	1 (8%)	2 (18%)	0 (0%)
Specified neo-adjuvant chem otherapy	10 (30%)	2 (17%)	3 (27%)	5 (50%)
Specified adjuvant chemotherapy	9 (27%)	4 (33%)	3 (27%)	2 (20%)
Specified neo-adjuvant radiation	4 (12%)	0 (0%)	2 (18%)	2 (20%)
Specified adjuvant radiation	7 (21%)	5 (42%)	2 (18%)	0 (0%)
Surgery in combination with any other therapy (e.g., hormone therapy, immunotherapy, undefined targeted therapy)	12 (36%)	6 (50%)	4 (36%)	2 (20%)
Chemotherapy	26 (79%)	9 (75%)	9 (82%)	8 (80%)
Radiation	16 (48%)	5 (42%)	8 (72%)	4 (40%)
Immunotherapy	6 (18%)	1 (8%)	2 (18%)	2 (20%)
Hormone therapy	2 (6%)	2 (17%)	-	-
Other unspecified targeted therapy	4 (12%)	3 (25%)	1 (9%)	-
Currently receiving treatment, *n* (%)				
Yes	12 (36%)	7 (58%)	3 (27%)	2 (20%)
No	20 (61%)	5 (42%)	8 (73%)	6 (60%)
Province, ^1^ *n* (%)				
Atlantic	7 (21%)	-	4 (36%)	3 (30%)
Central	15 (45%)	8 (67%)	3 (27%)	4 (40%)
Prairie	4 (12%)	2 (17%)	2 (18%)	-
Pacific	5 (15%)	2 (17%)	2 (18%)	1 (10%)
North	-	-	-	-
Race/ethnicity, ^2^ *n* (%)				
White	28 (85%)	11 (92%)	10 (90%)	7 (70%)
Black	-	-	-	-
Latin American	3 (9%)	1 (8%)	-	2 (20%)
Asian	2 (6%)	-	2 (18%)	-
Other	1 (3%)	1 (8%)	-	-
Level of education, *n* (%)				
Did not complete secondary school	2 (6%)	1 (8%)	1 (9%)	-
High school diploma or equivalent	4 (12%)	-	2 (18%)	2 (20%)
Undergrad or college certification	13 (39%)	6 (50%)	4 (36%)	3 (30%)
Post-graduate degree	12 (36%)	5 (42%)	4 (36%)	3 (30%)
Employment status, *n* (%)				
Working full-time	13 (39%)	7 (58%)	2 (18%)	4 (40%)
Working part-time	5 (15%)	1 (8%)	1 (9%)	3 (30%)
Retired	7 (21%)	1 (8%)	5 (45%)	1 (10%)
Not working due to cancer	3 (9%)	1 (8%)	2 (18%)	-
Not working for other reasons	3 (9%)	2 (17%)	1 (9%)	-
Other	2 (6%)	-	-	2 (20%)

Demographic data collection was guided by Statistics Canada categories [32]. ^1^ Atlantic encompassed Nova Scotia, Prince Edward Island, New Brunswick, Newfoundland and Labrador; Central encompassed Ontario and Quebec; Prairie encompassed Manitoba, Saskatchewan, and Alberta; Pacific encompassed British Columbia; North encompassed Nunavut, the Yukon, and the Northwest Territories. ^2^ Participants were able to select all that applied. Percentages sum up to more than 100%. ^3^ Data not reported by two participants with GI (liver) cancer; the GI cancer group included 6 participants with liver cancer, 3 with gastric cancer, and 1 with esophageal cancer. Abbreviations: GI, gastrointestinal; yrs, years.

**Table 2 cancers-17-03260-t002:** Participant quotations to illustrate perspectives on OS as a trial endpoint.

*Perspectives on OS as a Trial Endpoint*	
OS provides valuable information	“It helps you plan for the rest of your life. If somebody says to you, ‘You have one year to still be alive’ or ‘You have 10 years to be alive’ I think you would behave differently. You would pack these 10 years in one year. [laughs] And do everything you wanted to do … [OS] has some importance for the quality of life and what you want to achieve in your life.”—ID11, lung cancer
OS is easily understood	“… it’s information that we’ve been conditioned to want to know. Everyone wants to know, with a diagnosis, ‘Well, what does that mean for me? In terms of longevity, … what am I dealing with here?”—ID02, breast cancer
Survival is the main goal of treatment	“Yes … That was a biggie for me, just to survive as long as possible … do anything you can to make sure that you survive.”—ID22, lung cancer
Concerns about the statistical relevance of OS to an individual’s life	“Also, there’s just that random factor of all the other things that could lead to death. That, to me, feels like a huge variable that is not at all controlled. Anyone could die at any point in time for any measurable thing, but knowing that the cancer isn’t coming back means that that is one thing that I can eliminate from the everything else in life that I need to stress about.”—ID08, breast cancer
	“Give me the average, and the average doesn’t necessarily pertain to me anyway, right, especially with lung cancer. I mean, I know that there’s more and more non-smokers. There’s more and more younger people, but at the beginning, it was mostly older people who were getting it. I think they do still have a big say in our average.”—ID22, lung cancer
OS may take a long time to measure	“Just because of the length, it means that we’re still evaluating things that came out 10+ years ago. That scares me that if we’re waiting on certain metrics of overall survival that drugs that can save people’s lives aren’t going to be getting into people’s bodies in time.”—ID23, lung cancer

Abbreviations: OS, overall survival.

**Table 3 cancers-17-03260-t003:** Participant quotations to illustrate perceived strengths and drawbacks to the use of non-OS endpoints.

*Strengths of Non-OS Endpoints*	
Align with participants’ treatment goals	“To a cancer patient, clear margins are extremely important for your [physical] health and mental health … because if you can say, ‘Okay, there’s no more cancer in me,’ that gives you a very positive outlook on the future. So, yes, if a doctor offered me another treatment … that would say, ‘80% of people did very well with this. They had clear margins afterwards,’ yeah, I absolutely would.”—ID14, lung cancer
	“I wanted … treatment to essentially make me disease-free; … [so] I don’t have to go through the same thing in a few years time … sometimes when you’ve just climbed that mountain and there’s another peak which you have to climb again, it’s just like, ‘Not again. I can’t do this again.’ … So, yeah, that was very important for me to … kind of anticipate what’s in the future for me when it comes to this disease.”—ID29, GI (liver) cancer
Non-OS endpoints represent meaningful constructs	“I like [the] DFS [endpoint]. I don’t want to think about it in terms of a decade or 15–20 years. I really want to focus on something more achievable and relevant to me in the short-term.”—ID25, GI (liver) cancer
	“Just knowing that there is nothing there after the surgery, I guess. I wasn’t thinking in terms of a long life; I was thinking in terms of getting through a few years.” ID36, GI (esophageal) cancer.—ID36, GI (esophageal) cancer
	“I’m not going to have to worry about being off work and finances, barring other things in life. I’ll be around for my kids now that they’re older, and hopefully I can see them have kids. And I can think about retirement, all those types of things. Even though the overall might not be any different, I know that ‘Okay, but the next 10 years should look awesome.’”—ID18, breast cancer
Provide valuable additional clinical utility	“I am most interested in [RFS] because it weighs very heavily on me, and many cancer patients live with that sort of threat or fear for the rest of their lives, of recurrence … Knowing the rates of recurrence-free survival would be very useful information for me.”—ID02, breast cancer
	“100% [pCR] was an important goal for me.. Literally every single study identified complete pathologic response as the deciding factor of what your prognosis was going to be. It was really important for me to know and understand what that meant. So … it was the best case outcome, but especially for me with triple negative breast cancer, the prognosis and duration of disease-free survival is astronomically different whether you get a complete pathologic response or not.”—ID08, breast cancer
Access to treatment	“Definitely I would say to decision makers that, at the speed at which the science is going, if you’re waiting, as you wait, the cost is lives of people. And yes, there is uncertainty [with e.g., DFS], and I would put some boundary of certainty in the goodness and the challenges of treatment. But waiting for a very long time to make drugs appear, as somebody who has cancer, it’s kind of a little bit crazy [laughs].”—ID11, lung cancer
	“I think that our system, drug approvals and clinical trials take way too long to be able to get approved. It’s hard to see things that are available to people in other countries that should be available to us here, but you can’t have access unless you have money. It’s frustrating.”—ID07, breast cancer
*Perceived drawbacks to non-OS endpoints*	
Less information on long-term treatment impacts	“There are some long-term use impacts associated with a drug that are just unknown at the time of [non-OS endpoints] that only in time would become evident. And so that is a risk that you would have to consider before agreeing to the treatment. But I think in most circumstances, and certainly in my circumstance, the risk of the cancer killing you is greater, so you kind of take those risks and hope for the best.”—ID02, breast cancer
	“Well, faster access would mean less information and less reassurance to offer people because there wouldn’t be the data.”—ID07, breast cancer
May not meet all patient expectations	“I think one of the biggest ones (drawback of treatment measured to the standard of a non-OS endpoint) is false hope, right? If there is this mindset that it’s probably going to lead to longer overall survival and people go into that expecting it, and then it turns out it’s not the case, I think that, emotionally, psychologically that would be very hard on patients. I mean, I guess that would probably be the biggest one, especially if the quality of life is that much harder. You put your body on the line in the hopes of having these benefits, and if they don’t materialize the way you expect them to, yeah, I just wonder how much people would suffer, not just physically with symptoms they weren’t expecting but emotionally for not getting the benefits they were hoping for.”—ID23, lung cancer
	“Overall survival would be much clearer in my mind and easier to understand than an event-free survival prediction, I think … it’s not a bad thing. I just think it’s not quite as clear, it wouldn’t be as clear in my mind what that meant exactly, whereas “survival” survival, you sort of understand that pretty basically.”—ID37, GI (stomach) cancer

Abbreviations: DFS, disease-free survival; EFS, event-free survival; GI, gastrointestinal; OS, overall survival; pCR, pathological complete response; RFS, recurrence-free survival.

**Table 4 cancers-17-03260-t004:** Participant quotations to illustrate other factors influencing perspectives on the importance of non-OS endpoints, and key messages from participants about the value of non-OS endpoints for decision-making.

*Valuing specific non-OS endpoints differently*	
Assumed a better pCR would translate to longer OS, RFS, DFS or EFS—and in turn, longer OS—even when told there was no evidence supportive of this link	“Yeah, because I’m pretty sure if I keep getting the same disease and keep getting more setbacks in my health, then it would also decrease my overall survival. So, I think there’s a correlation between being disease-free and my overall survival. I think if you achieve that first step, then your overall survival also increases.”—ID29, liver cancer
	“I believe the pathological complete response gave me a chance that I am cancer free. Which means I will … overall survival time will be very long, right.”—ID24, lung cancer
The hope provided by a pCR or the assurance of a long DFS or RFS time is an important part of the appeal of non-OS endpoints for the participants.	“So I think [pCR] does give that hope that at least targeting the cancer that it’s meant to treat, it’s successful, right? And so it is kind of a sign in the right direction.”—ID06, breast cancer
	“But if a certain treatment gives a very high incidence of recurrence-free survival, like you hope you land in that percentage, [laughs] and having that to combat some of the trauma and fear of being a cancer patient and the long-term impacts of that, yeah, I’d really like to know that information.”—ID02, breast cancer
*Other factors influencing perspectives on the importance of non-OS endpoints in early-stage cancers*	
Younger participants or participants with better prognoses saw a cure as possible and having treatment options that provided a better HRQoL as a primary goal	“If the side effects of the treatment were impeding my lifestyle or my overall wellbeing … Say, for instance, they want me to go on tamoxifen, and if I’m going to have decreased libido and I’m going to be dry and I’m going to be—I don’t know—gaining 50 lbs. and … increasing my chances of getting cervical cancer, I would probably say no. Yeah, so if the side effects are going to be worse than what I am right now, then I would probably not take that extra medication.”—ID10, age 41, breast cancer
	“Obviously, try to remove the cancer and be cancer-free because every time you think of cancer, you think of chemotherapy. You think of being nauseous. You think of being sick. So, I think the quality of life as you’re going through these treatments is very important. Obviously, at different stages of the cancer, it needs to be treated more aggressively, of course. I’m just lucky I was at that stage where I could be treated in a not so aggressive.”—ID27, stage A, GI (liver) cancer
For those who had or wanted to have families, treatment goals often factored this into consideration.	“I know many, many other young women who have gone through breast cancer, especially those who had children, they were willing to do everything-like, throw the kitchen sink—and it did not matter what the studies showed. If you had equal chance of survival doing a double mastectomy versus a lumpectomy, they chose the double mastectomy even though there was no greater chance of survival, but feeling like they did everything that they could to feel like they had done everything to survive.”—ID01, breast cancer
Participants with poorer prognoses valued survival and having treatment options for subsequent lines of treatment	“I think if you’re talking to someone who has a high risk of recurrence or was diagnosed stage 4 de novo or has had a recurrence, I think the perspective changes. It’s not about not having the cancer come back. It’s all about ‘how long can I live?’ So, I think it really depends on the diagnosis.”—ID08, breast cancer
	“Yes. If I were a cancer patient that my percentage wasn’t very good of surviving, I would definitely take [a drug that has outcomes expressed as event-free survival not overall survival]. I think it depends on where you are in your cancer journey.”—ID14, lung cancer
For patients who had experienced relapse or never achieved ‘no evidence of disease’, survival was still a priority, but individuals often stressed the importance of maintaining HRQoL in the face of chronic disease.	“And so definitely seven years ago, I was thinking, ‘Yes, it’s on or off,’ but now I think I’ve resolved myself that it might be there for long term, and as long as there are medication to take care of it, then it’s fine.”—ID11, lung cancer
The treatment options available for the type and stage of cancer influenced perspectives on risk and innovation in treatments	“I think also the type of cancer that you have and what other treatment options are available. Is this the only treatment that is out there, or the only option, or is it the best chance or is it in combination with something else?”—ID06, breast cancer
	“Yes. If I were a cancer patient that my percentage wasn’t very good of surviving, I would definitely take [a treatment that reported outcomes in terms of EFS but not OS]. I think it depends on where you are in your cancer journey.”—ID14, lung cancer
	“Just knowing that there is [no treatment option] there after the surgery, I guess, I wasn’t thinking in terms of a long life, I was thinking in terms of getting through a few years.”—ID36, GI (gastric) cancer
Participants believed that there was a point during treatment, especially for older adults, when HRQoL would become a priority over survival	“I think it depends on the age of the person, honestly. Because if I’m an 80-year-old lady or person and you’re saying like, ‘You can live 10 more years, but those are going to be crappy years because you’re going to be fighting,’ like ‘you’re going to have to keep coming back to the hospital and dealing with that kind of stuff,’ I don’t know if I would choose to do the treatment. But like at 29, I would do anything.”—ID05, breast cancer
	“I think knowing I was told I was cancer-free would be more important than anything. I am 65 years old, at the time 64 when I was told I was cancer-free, just being told I am cancer-free is most important”—ID36, GI (esophageal) cancer
Even among those who felt DFS, RFS, and/or EFS strongly aligned with their treatment values, they remained concerned about side effects that “could be worse than the disease” or could negatively impact their lifestyle.	“Well, depending on the side effects of the treatment, what was involved in that type of thing. But if it was reasonable side effects and without too much change in your lifestyle compared to the traditional treatments, if it was a new treatment that had similar effects on you during the time that you’re being treated, if you had a better chance of a pathological complete response, I would certainly accept that.”—ID37, GI (gastric) cancer
*Patient agency and contributions towards decision-making*	
Participants revealed that they do not always feel empowered to advocate (or were successful in advocating) for themselves and their treatment goals	“I guess [my treatment goals changed to be] overall effects, the side effects. Like I said, my joint pain is terrible. They did not tell me that I could have heart issues, which … I’ve now had two surgeries”—ID04, breast cancer
	“I would’ve really looked at the research and talked to people that maybe had went through it and what their opinions were, also taking into consideration what my oncologist is saying, but I probably would’ve made different choices.”—ID04, breast cancer
	“Yeah, I think, over time … that I learned more through my process of how to advocate, and I started doing that, actually, luckily, fairly early on in the process. I, behind the scenes, help cancer patients advocate today.”—ID20, lung cancer
Many participants felt that clinicians should be making the ultimate treatment decisions for the patients	“I would want to have availability to everything within like basic safety concerns. As long as I’m reassured by competent doctors that this is safe, better, I would want to have access to new treatments.”—ID25, GI (liver) cancer
	“My initial … I guess my initial hopes and thoughts were that I would be presented with sort of the cutting edge, with what was the most up to date and most promising. And that would be presented by a team of experts in my particular cancer. That was my hope and my expectation.”—ID21, lung cancer
*Messages to decision-makers*	
Patients and clinicians should be given more treatment options by population-level decision makers	“And again, I think patients are good or getting better, or we need to allow them the space to be their own advocates and understand the risks, and if they want access to something because it’s the only thing that might work for them, that they have the right to try it, and that it’s not held up in a bureaucratic process and that they pass away in that time is … That shouldn’t be what’s happening in our system.”—ID06, breast cancer
A desire for decision makers to consider non-OS endpoints when making decisions about access for treatments for early-stage cancers	“Especially in a country like Canada where health care policy is trying to set for huge populations at a time, I really think there has to be more consideration for individuals’ preferences. So, if decision makers say, ‘We’re going to value overall survival more than disease-free survival,’ or vice versa, they’re making a value decision for millions of people who might feel differently. I get quite upset with this concept of the standard of care, of these boxes that people get put into because they have the best outcomes. Well, what are they counting as outcomes? What are they measuring? Those are all decisions that are subjective.”—ID23, lung cancer
	“Make it available and that way we can have that information,” and you can have the information, or the doctors, so they can move on with [the treatment] or not.”—ID38, GI (gastric) cancer
Timely and equitable access to treatments saves lives	“I kept saying, throughout this whole thing that I went through, ‘Good Lord, I would not be alive if I got this cancer 20-30 years ago. Thank goodness there’s new medicines out there’” and that pretty well answers my question.”—ID17, lung cancer
	“I can’t imagine them waiting for overall survival to put something on the market. Then I find it amazing how many people are saying, ‘Well, the drug is available in the states, but it’s not available here yet,’ or, ‘It’s available in Ontario, but it’s not available yet in New Brunswick. It’s not covered by our insurance yet, but its covered by so-and-so’s insurance.’ There are so many things out there that … oh, yeah. It’s a tough one, but overall survival, I don’t think they can wait for that state. They just have to do it. You have to save as many of us as possible.”—ID22, lung cancer

Abbreviations: DFS, disease-free survival; EFS, event-free survival; GI, gastrointestinal; HRQoL, health-related quality of life; OS, overall survival; pCR, pathological complete response; RFS, recurrence-free survival.

## Data Availability

The data that support the findings of this study are available on request from the corresponding author. The data are not publicly available due to privacy or ethical restrictions.

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
