# Peer review of "“Are You Just Looking to ‘Survive’?”: A Qualitative Study of Importance of Oncology Endpoints Beyond Overall Survival in Early-Stage Cancer [Author-notes fn1-cancers-17-03260]"

_cancers, 2025, doi:10.3390/cancers17193260_

Round 1

Reviewer 1 Report

Comments and Suggestions for Authors

The study described in this article was conducted among 33 people treated for breast, lung, and gastrointestinal cancers in Canada. Participants provided perspectives on OS, RFS, disease-free survival (DFS), EFS, and pathological complete response (pCR) endpoints. The authors confirmed the relevance and importance of non-OS endpoints for Canadians with early-stage cancer and emphasized the participants' desire for rapid acceptance of treatments with proven improvement in non-OS endpoints. The study and its results are worth publishing. They fill a gap in the literature on this topic.

The introduction provides a comprehensive introduction to the topic addressed by the authors. The objectives are well-defined and presented at the end of the introduction. The materials and methods are described in detail, allowing for the reconstruction of the study. Results: The results are described accurately and presented in tables, figures, and text. The article is enriched by the opinions of patients expressed during the study, which are cited. The discussion describes the most important results obtained and compares them with those of other studies. The study's limitations are described. Conclusions were drawn appropriately based on the obtained results.

I have a few minor comments:

Materials and methods: Study inclusion criteria have been provided. It is also worth providing exclusion criteria. There is no information regarding the year in which the study received ethical approval from the WCG IRB. There is no information regarding the software used to perform the statistical analyses (country, license).

Results: Figure 1 is in small print and will be illegible for readers. It may be worthwhile to provide some of the information in text within the main text of the article, or to increase the font size of the figure.

Discussion: It is worth adding a paragraph at the end of the discussion about future implications.

Minor punctuation errors in the article and references (spaces, periods, commas, missing the bolded year of publication).

Author Response

The study described in this article was conducted among 33 people treated for breast, lung, and gastrointestinal cancers in Canada. Participants provided perspectives on OS, RFS, disease-free survival (DFS), EFS, and pathological complete response (pCR) endpoints. The authors confirmed the relevance and importance of non-OS endpoints for Canadians with early-stage cancer and emphasized the participants' desire for rapid acceptance of treatments with proven improvement in non-OS endpoints. The study and its results are worth publishing. They fill a gap in the literature on this topic.

The introduction provides a comprehensive introduction to the topic addressed by the authors. The objectives are well-defined and presented at the end of the introduction. The materials and methods are described in detail, allowing for the reconstruction of the study. Results: The results are described accurately and presented in tables, figures, and text. The article is enriched by the opinions of patients expressed during the study, which are cited. The discussion describes the most important results obtained and compares them with those of other studies. The study's limitations are described. Conclusions were drawn appropriately based on the obtained results.

I have a few minor comments:

  1. Materials and methods: Study inclusion criteria have been provided. It is also worth providing exclusion criteria.

As suggested, we have updated the study design section to include study exclusion criteria.

  1. There is no information regarding the year in which the study received ethical approval from the WCG IRB.

We have added details on the month and year (January 2025) we obtained ethical approval to our ethics section.

  1. There is no information regarding the software used to perform the statistical analyses (country, license).

Thank you for your comment. We have inserted information regarding the version of the software we used and its originating country.

  1. Results: Figure 1 is in small print and will be illegible for readers. It may be worthwhile to provide some of the information in text within the main text of the article, or to increase the font size of the figure.

We have increased the font size for Figure 1 and removed the quotes. We have instead moved these to a table for ease of reading and consistency of formatting with the other quotes presented.

  1. Discussion: It is worth adding a paragraph at the end of the discussion about future implications.

We agree that including future implications strengthens the manuscript, and we have added key considerations for future research to the last paragraph of the Discussion.

  1. Minor punctuation errors in the article and references (spaces, periods, commas, missing the bolded year of publication).

Thank you, we have reviewed the document in full for formatting errors.

Reviewer 2 Report

Comments and Suggestions for Authors

This is an interesting study reporting the perspectives of Canadian cancer patients on the value of various endpoints typically used in cancer trials. Although the study is both interesting and important, I believe the data presentation could be improved.

  1. The Introduction should include a passage describing the need to investigate patient views. From this perspective, the first paragraph of the Discussion section could be relocated to the Introduction. In its current format, the study objective appears somewhat surprising, as no preceding information suggests what the objective might be.
  2. The study objective does not align well with the methodology. The authors state that 'the objective of this study was to explore the perspectives of Canadians with early-stage cancer on the perceived value of non-OS endpoints,' yet Table 1 shows that only 10 out of 33 patients had stage I disease. Furthermore, the last paragraph of the Discussion notes that 'additional strengths include the diversity of experience and perspectives offered by including participants with different cancer types and disease stages.' Therefore, the authors should reconsider the formulation of the study objective presented in the Introduction and Conclusion sections.
  3. Given that 12 of 33 participants held postgraduate degrees, the sample may not adequately reflect the views of individuals with lower educational attainment. This potential bias should be acknowledged as a study limitation.
  4. It is not clear from the description of the study methodology whether the participants were familiarized with the indicators discussed. Please provide more detail on this point.
  5. Please explain what types of analyses were carried-out via Excel

Author Response

Thank you very much for your review of our manuscript. We have outlined our responses to each of your comments below. We appreciate the opportunity to have revised the manuscript and hope you find these changes to have improved the manuscript.

Kind regards

Shelagh Szabo

Reviewer #2

This is an interesting study reporting the perspectives of Canadian cancer patients on the value of various endpoints typically used in cancer trials. Although the study is both interesting and important, I believe the data presentation could be improved.

  1. The Introduction should include a passage describing the need to investigate patient views. From this perspective, the first paragraph of the Discussion section could be relocated to the Introduction. In its current format, the study objective appears somewhat surprising, as no preceding information suggests what the objective might be.

Thank you for highlighting this point. We agree that the Introduction was missing a clear statement to position the focus of our study within the broader landscape of the field. We have moved the relevant content from paragraph 1 of the discussion to better frame the study objective and ensure it is anticipated by the reader.

  1. The study objective does not align well with the methodology. The authors state that 'the objective of this study was to explore the perspectives of Canadians with early-stage cancer on the perceived value of non-OS endpoints,' yet Table 1 shows that only 10 out of 33 patients had stage I disease. Furthermore, the last paragraph of the Discussion notes that 'additional strengths include the diversity of experience and perspectives offered by including participants with different cancer types and disease stages.' Therefore, the authors should reconsider the formulation of the study objective presented in the Introduction and Conclusion sections.

We can see how our use of the term ‘early-stage’ could be confusing, as the implications of what is considered early-stage often vary across cancer types.  Our goal was not to recruit within a strict clinical definition of early-stage, but rather to explore the perspectives of people who were treated for cancers which often have a better prognosis and therefore have treatment regimens that align with treating with curative intent (as opposed to those with metastatic cancers who likely have different perspectives). This choice was guided by the opinion of the study team’s clinical experts and the published literature for these cancer types. To provide additional clarity, we have been more explicit that we recruited participants who had early-stage cancer or ‘were treated with curative intent’, to better delineate the population.  We have provided this clarification in the abstract, added this clarification in the recruitment methods, discussed this choice in the discussion section, and added clarification in the conclusion paragraph.

  1. Given that 12 of 33 participants held postgraduate degrees, the sample may not adequately reflect the views of individuals with lower educational attainment. This potential bias should be acknowledged as a study limitation.

Thank you, as suggested, we have incorporated this potential bias into the limitations section of the Discussion.

  1. It is not clear from the description of the study methodology whether the participants were familiarized with the indicators discussed. Please provide more detail on this point.

We have now expanded on our summary of the process taken to familiarize participants with the endpoints within the Data Collection section.

  1. Please explain what types of analyses were carried-out via Excel

We have added details on how both NVivo and Excel were used in the Analysis section.

Reviewer 3 Report

Comments and Suggestions for Authors

The manuscript « “ARE YOU JUST LOOKING TO ‘SURVIVE’?”: A QUALITATIVE STUDY OF IMPORTANCE OF ONCOLOGY ENDPOINTS BEYOND OVERALL SURVIVAL IN EARLYSTAGE CANCER» is a qualitative study examining how people with early-stage cancer in Canada value outcomes measured in clinical trials.

The manuscript can be accepted after major revision.

The article looks interesting and quite promising.

1) Precise analysis data obtained as a result of the work must be provided. Heat maps or correlation diagrams should be added.

2) Diagram 1 needs to be redone: the small font makes it impossible to understand what is written and to determine the presence or absence of correlation. For example, is there a correlation between cancer types, treatment types, stages, etc.?

3) The article does not sufficiently discuss the results and their comparison with previously obtained results.

4) In addition, the conclusion needs to be rewritten.

Author Response

Thank you very much for your review of our manuscript. We have outlined our responses to each of your comments below. We appreciate the opportunity to have revised the manuscript and hope you find these changes to have improved the manuscript.

Kind regards

Shelagh Szabo

Reviewer #3

The manuscript « “ARE YOU JUST LOOKING TO ‘SURVIVE’?”: A QUALITATIVE STUDY OF IMPORTANCE OF ONCOLOGY ENDPOINTS BEYOND OVERALL SURVIVAL IN EARLYSTAGE CANCER» is a qualitative study examining how people with early-stage cancer in Canada value outcomes measured in clinical trials.

The manuscript can be accepted after major revision.

The article looks interesting and quite promising.

  1. Precise analysis data obtained as a result of the work must be provided. Heat maps or correlation diagrams should be added.

Thank you for this insight on additional potential avenues of analysis for this study. Unlike in typical quantitative analyses that focus on measuring associations, the qualitative analyses described here focus on understanding patterns in the data and the contexts underlying them. Reflexive thematic analysis in particular aims to develop in-depth understandings of the experiences of participants and their thoughts and feelings through iterative engagement with participants’ narratives, and in this analysis, we use contextual factors to inform our interpretations of patterns we see in the data. To your point raised here and in the comment directly below, we recognize that how the data were initially presented in Figure 1 fails to highlight the relationships between preferences for endpoints and the clinical and sociodemographic characteristics that define the sample. Based on the comment in point 2 below, we have reworked Figure 1 to move the quotations on these topics to a new Table 3, to make the discussion of factors influencing perspectives on endpoints more clear (which are also referenced in Table 2). Should the reviewer and editor think that an additional figure clarifying these relationships would have value we would be happy to develop this; which we did not include at this time due to manuscript length.

  1. Diagram 1 needs to be redone: the small font makes it impossible to understand what is written and to determine the presence or absence of correlation. For example, is there a correlation between cancer types, treatment types, stages, etc.?

We have increased the font size for Figure 1, and added the quotes to a table for ease of reading and consistency of formatting with the other quotes presented. Please see our response to comment 1, which we hope also addresses the second part of this comment.

  1. The article does not sufficiently discuss the results and their comparison with previously obtained results.

Thank you for your suggestion. We have added additional details to Discussion paragraphs 2 and 6, highlighting additional studies previously conducted that are relevant to this work.

  1. In addition, the conclusion needs to be rewritten.

Thank you for this note. We have re-worded the Conclusion to emphasize clarity of ideas and to better highlight the key take-aways of the study through a qualitative lens.

Round 2

Reviewer 2 Report

Comments and Suggestions for Authors

The authors addressed all comments raised during the previous round of revision. 

Reviewer 3 Report

Comments and Suggestions for Authors

The manuscript can be accepted in its current form.